# Quantitative Image Processing for Three-Dimensional Episcopic Images of Biological Structures: Current State and Future Directions

**DOI:** 10.3390/biomedicines11030909

**Published:** 2023-03-15

**Authors:** Natalie Aroha Holroyd, Claire Walsh, Lucie Gourmet, Simon Walker-Samuel

**Affiliations:** 1Centre for Computational Medicine, University College London, London WC1E 6DD, UK; 2Department of Mechanical Engineering, University College London, London WC1E 7JE, UK

**Keywords:** episcopic, imaging, microscopy, annotation, quantification

## Abstract

Episcopic imaging using techniques such as High Resolution Episcopic Microscopy (HREM) and its variants, allows biological samples to be visualized in three dimensions over a large field of view. Quantitative analysis of episcopic image data is undertaken using a range of methods. In this systematic review, we look at trends in quantitative analysis of episcopic images and discuss avenues for further research. Papers published between 2011 and 2022 were analyzed for details about quantitative analysis approaches, methods of image annotation and choice of image processing software. It is shown that quantitative processing is becoming more common in episcopic microscopy and that manual annotation is the predominant method of image analysis. Our meta-analysis highlights where tools and methods require further development in this field, and we discuss what this means for the future of quantitative episcopic imaging, as well as how annotation and quantification may be automated and standardized across the field.

## 1. Introduction

Episcopic microscopy refers to imaging techniques that utilize serial sectioning and block-face imaging to create high-resolution large-volume three-dimensional (3D) datasets. High Resolution Episcopic Microscopy (HREM) and its associated techniques (namely, Optical HREM, Episcopic fluorescent image capture (EFIC) and multi-fluorescent HREM (MF-HREM) (Figure 1A)), require samples to be embedded in a supporting material to enable automated sectioning at the micron scale. After each section is removed, an image is captured of the cut surface, enabling a 3D image stack to be constructed (Figure 1B). In contrast to serial histology, the image stack produced by episcopic microscopy is inherently aligned and therefore requires no registration step. Furthermore, unlike intact-tissue imaging techniques such as light sheet and confocal microscopy, episcopic imaging is not limited by light penetration or microscope working distance and does not require samples to undergo clearing. In short, these methods allow large tissue volumes (up to 1 cm^3^) to be imaged in an automated fashion, while achieving resolution down to the cellular level. As such there are of interest to many fields of biological imaging.

Each of the episcopic techniques above have established experimental protocols across a range of biological samples; and, in the case of optical HREM and EFIC, have been used to test biological and medical hypotheses for over a decade [1]. Alongside the development of experimental protocols, there has been an associated development of image processing tools to extract quantitative information from HREM images. Much of the quantitative analysis of HREM images to date has relied upon manual annotation of structures of interest, for example, to extract tissues volumes, surface areas or number of anomalies. As HREM can be a high-throughput imaging modality (particularly for mouse embryo studies), and the datasets produced may be very large, manual image processing can present a bottleneck both in terms of both computational resource and researcher time.

In this review we outline the goals and methods of quantitative 3D image processing and present results from a systematic review into the current practices for image processing approaches in the episcopic imaging literature. Additionally, we highlight obstacles preventing wider use of quantitative image analysis. Finally, we propose avenues for further research and suggest what future developments we may expect.

**Figure 1 biomedicines-11-00909-f001:**
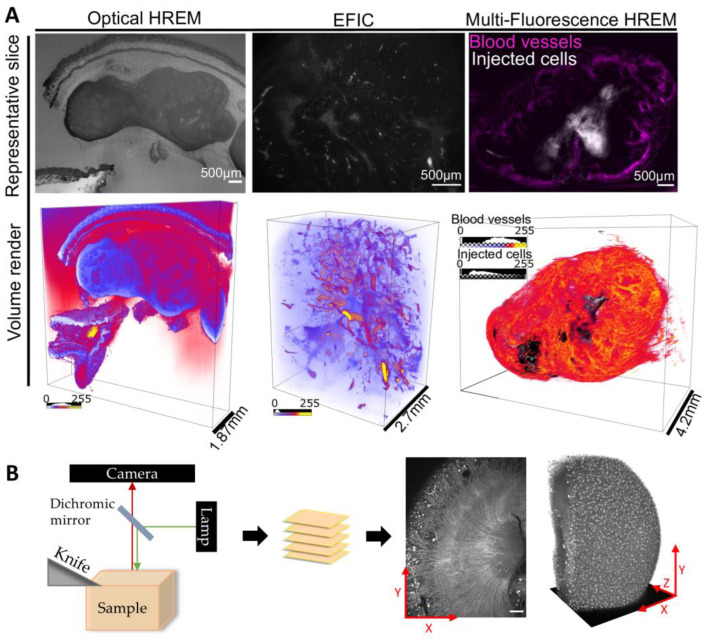
(**A**) Representative 2D slices are shown to illustrate a range of sources of contrast used in episcopic imaging: Optical HREM (eosin), EFIC (autofluorescence) and MR-HREM (targeted fluorescence labels). Reprinted with permission from Ref. [2]. All three panels show a subcutaneous murine tumor model. Different features are visible depending on the contrast source used. (**B**) A schematic illustration of an episcopic imaging system. A fluorescence microscope is positioned above a mechanized sample stage with inbuilt microtome blade. Sequential 2D images are captured of the block surface as the top layer of the sample is cut away the blade. The resultant stack of images is then reconstructed to create a 3D volume.

### 1.1. Background on Episcopic Imaging Techniques

HREM was originally developed as a high-throughput platform for phenotyping transgenic mouse models [1,3,4] and has since been developed into a commercial system (Optical HREM, Indigo Scientific). For these embryo phenotypic studies eosin was added to the embedding medium, with image contrast resulting from eosin blocking (when bound to eosinophilic proteins, the fluorescence of unbound eosin is inhibited or frequency-shifted [5]). This produces images with an appearance similar to the inverse of a traditional eosin staining in histology. Similarly, imaging of LacZ expression through x-gal staining has also been used in conjunction with eosin-HREM, using a secondary filter set. As with Eosin contrast, x-gal staining blocks the native fluorescence of the resin embedded sample. 

Episcopic Fluorescence Image Capture (EFIC) is a variant of HREM in which samples are embedded in non-fluorescent resin and contrast is provided solely by the native autofluorescence of the tissue [6]. More recently MF-HREM was developed to enable imaging of multiple targeted fluorescent stains in a similar manner to other 3D optical imaging modalities such as light sheet or serial two-photon microscopy [2]. 

Variants of episcopic imaging have been developed that utilise cryosectioning to circumvent some of the difficulties of standard HREM sample preparation [7,8]. Rather than dehydrating and embedding samples in resin, cryosectioning episcopic microscopy requires that the samples are embedded in OCT (Optical cutting temperature) compound at low temperature to harden them, enable accurate sectioning. This can be more time efficient and compatible with a greater number of fluorescent stains. 

Applications of HREM and its variants have included large-scale embryo phenotyping to build up an atlas of embryonic-lethal gene knockouts `Deciphering the Mechanisms of Developmental Disorders (DMDD)’ [4], assessing blood vessel morphology in human skin biopsies [9,10,11], in addition to other use cases where a large 3D field of view is essential.

### 1.2. The Importance of Quantitative Image Analysis 

Quantitative image analysis is required biological hypotheses to be rigorously tested and results to be reported with confidence. Moreover, quantification can be used to extract more information from image data and, when automated, facilitate high throughput analysis. Well established quantification pipelines allow researchers to standardise their image analysis, and the objectivity of quantitative analysis makes for more robust interpretations of data. Finally, quantised results can be used for further experiments such as in-silico modelling or training of a machine learning model. However, performing quantitative analysis on the large 3D data sets typically produced by episcopic imaging is a non-trivial task that requires specialised approaches depending on the data type and the aims of the analysis. 

### 1.3. Methods of Quantitative Imaging 

There are a few key methods for quantification that are relevant to episcopic imaging data analysis; as the pixel intensities of episcopic images do not correlate with biological properties (unlike techniques such as Bioluminescence Microscopy), analysis is largely dependent on image annotation. Annotation can take the form of object detection or segmentation. Object detection usually takes the form of a bounding box or coordinates to identify the feature of interest, while segmentation labels every pixel within the feature as belonging to that feature (where this is done within the feature of interest only, it is referred to as instance segmentation; semantic segmentation requires every pixel in the image to be assigned to a class, e.g., vessel or background) (Figure 2). Object detection is generally easier to achieve, but segmentation is more information-dense. Once an image is annotated, further analyses can be performed, for example object counting, distribution and morphometry of objects. 

Annotation can be performed manually, by tracing or highlighting objects, or by computational means. Widely used methods include pixel intensity-thresholding, filtering (for example, Frangi filtering for blood vessel enhancement [12]) and, more recently, deep learning. The processing time, labour and level of expertise required varies depending on the method. Fully manual segmentations performed by a highly trained expert are generally considered the gold standard, but are time consuming and can vary between operators. Semi-automated methods may require expertise to fine-tune, whilst fully automated methods do not require human intervention, making them most suited to batch analysis. The ideal image processing pipeline is one which is fully automated to facilitate high throughput, objective results that can be replicated by all users with the same software implementation. 

## 2. Methods

Literature searches were performed in Google Scholar using the term “high resolution episcopic microscopy” OR “episcopic fluorescence image capture” (case ignored) with a date range of January 2011 to December 2022. 463 matches were initially returned. Based on the title, abstract and methods for these results, review articles were excluded alongside articles which did not use HREM, EFIC or a related episcopic technique. This left 167 relevant articles. 

Subsequent exclusion criteria removed articles which did not use biological samples or were protocol descriptions only, without new experimental data or new image processing methods. Conference abstracts without enough detail to extract meta-analysis data were also excluded. From the remaining 123 articles, ten features were extracted: YearContrast typeStructure under investigationModel organismWas annotation performed?If yes: was this segmentation, object detection, or both?Was quantitative data presented?Method of annotation /quantification—organized into categories determined during analysisSoftware usedWas the raw image data deposited in an online repository?The type of publication (paper, conference paper, thesis)

## 3. Results

### 3.1. Trends in Prevalence of Quantification

Since 2011, episcopic microscopy data has featured in 123 publications, including 115 journal papers, three conference papers, two correspondences, two doctoral theses and one case report. A table summarizing all these publications can be found in the Appendix A). By far the most widely used contrast method has been eosin labelling (featured in 90 papers), sometimes combined with acridine orange or X-gal labelling (Figure 3A). EFIC was the second most used contrast method, being featured in 24 papers. Four papers used fluorescence labelling, each of which included segmentation of tissues and quantitative results. Three papers employed cryosectioning, of which two featured quantitative analysis. Data was made available online for 27 papers (22%), 17 of which were a part of the Deciphering Mechanisms of Developmental Disorders (DMDD) project [13].

The most commonly imaged species was mouse (85 papers), followed by human tissue (20 papers). There has been a strong focus on visualizing mouse embryos, in particular for assessing cardiac morphology (represented in 55 papers).

Over these past ten years, the number of papers published has averaged ten per year, with an increase in the last two years to 17 in 2021 and 16 in 2022. Of these papers, the proportion which provide quantitative image data has fluctuated over time (Figure 3B). In total, just over half of the papers published since 2011 (55%) provided quantitative results, but where annotation was performed that number increased to 70%, signifying the important role annotation plays in quantification. Furthermore, the approach used to annotate data has shifted, with image segmentation being favored over object detection since 2016. 

A quarter of the papers reviewed used episcopic imaging for visualization without annotation; often this data was presented alongside other experimental techniques such as in-vivo imaging, immunohistochemistry and genetic sequencing. For example, Dupays et al. [14] used HREM visualizations of mouse embryonic hearts to confirm morphological changes (ventricular wall thinning and septal defects) associated with low levels of transcription factor NKX2-5. This result was reported alongside transcriptome analysis and Chromatin Immunoprecipitation Sequencing (ChIP-Seq) to identify 309 genetic targets of NKX2-5 and indicate its role as a regulator of cardiac chamber development. The ability of epsciopic microscopy to capture organ-wide morphological changes in 3D makes in a valuable tool for such studies.

### 3.2. Trends in Methods of Quantification

Through the analysis of the last decade of HREM papers, we have identified seven broad categories of quantitative analysis being performed on episcopic image data. We define these categories as: Scoring/counting of abnormalities (as defined in an atlas)Object counting (no agreed ontology needed)Registration basedHand 2D measurements of anatomic features after reslicing data3D morphological analysis, for example: volumes, networksFractal analysis of heart (standardized method)Other (for example, orientation of fibers)

3D morphological measurements were the most commonly applied method of quantification, occurring in 25 papers and becoming more prevalent over time (Figure 4A,B). This morphological analysis was often performed in conjunction with other methods such as scoring or counting of abnormalities. For example, Geyer et al. [15] presented morphological measurements of mouse embryo vasculature alongside frequency of anatomical features and defects. This was used to thoroughly describe normal cardiovascular development and provide a baseline against which to compare abnormal phenotypes. However, a limiting factor for use of this method is the need for segmentation, which was completed by hand in over half of the papers reviewed here. Simple segmentations were employed in many of these papers to remove background and enable measurements of volumes, surfaces areas and distances, but more advanced morphological analysis requires segmentation of complex tissue structures. For example, Sweeney et al. [16] produced graphical representation of the vascular connectivity to facilitate network-wide analysis, requiring a full segmentation of the entire vasculature. This segmentation was produced semi-manually, using a region-growing tool to select connected regions according to pixel intensity (magic wand tool, Amira v2020.2) and manual correction. Producing such a segmentation is highly labor intensive so cannot practically be applied to large datasets.

Scoring or counting of abnormalities was the second most widely employed method, appearing in 19 papers, and notably increased in use from 2016 onwards. This is potentially associated with the publication a couple of years earlier of the Deciphering Mechanisms of Developmental Disorders (DMDD) database: an open-source archive of HREM data cataloguing phenotypical changes associated with different embryonic lethal gene knockouts [13]. As part of the DMDD project, two papers were published outlining detailed protocols for phenotyping embryos using HREM image data [17] and making use of the DMDD resource [18]. Seven papers analyzed here make specific reference to one of these protocol papers in their methods sections. This highlights the utility of open-source data platforms and widely available software for establishing a reproducible image quantification protocol. An advantage of scoring/ counting is that, as a method of analysis, it does not require segmentation, and thus can be performed by an expert observer in less time than a segmentation-based method. The vast majority of papers employing a scoring of counting method did so manually. The drawbacks of this method is that it may require a high degree of expertise to recognize abnormalities, and where there is a binary output for a given sample (e.g., normal versus abnormal) a large number of samples must be assessed to allow meaningful quantification. 

Seven papers included fractal analysis of the heart. This method produces a quantitative measure of complexity or surface roughness (fractal dimension, FD) and is wieldy used in cardiology [19]. Captur et al. [20] used fractal analysis of HREM images to quantify trabeculation in mouse models of hypertrophic cardiomyopathy, indicating abnormalities in the compaction of the myocardium. Additionally, the distribution of myocardial crypts was compared between mutants and wild types using the standardized American Heart Association bull’s-eye plot [21]. Standardized analysis allows these results to be easily interpreted and compared with results published elsewhere.

Other quantification methods began appearing in papers from 2017 onwards. Analysis of cell orientation was performed in three papers [2,22,23]. Carcia-Canadilla et al. [23] calculated the orientation of cardiomyocytes from HREM images using a structure tensor method developed in-house, and quantified the uniformity of myocyte orientation to demonstrate that disorganization of cardiomyocytes can be seen before birth in mouse models of hypertrophic cardiomyopathy. Similar methods were used to quantify white matter orientation by Walsh et al. [2], in this case employing the OrientationJ plugin of ImageJ. Two papers used HREM data as the basis for computational models: Sweeney et al. [16] developed a model of blood flow in microvascular networks, which was tested in the vascular network extracted from an MF-HREM image of the mouse medulla. Le Garrec et al. [24] created a 3D model of mouse heart morphogenesis with morphological measurements taken from HREM data used for parameterization; this model provided insights into the mechanisms involved in heart looping. While these examples are in the minority, it is likely that we will continue to see a wider variety of quantification methods being applied to episcopic data in the future.

### 3.3. Methods of Image Annotation

Manual annotation was the most common method for both object detection and segmentation. Overall, 76% of papers presenting quantitative image analysis relied, at least in part, on manual annotation (70% for papers employing segmentation, 89% for papers using object detection) (Figure 5B). Automated and semi-automated methods were less widely used: intensity thresholding was employed in six papers, while vasculature was segmented via Frangi filtering in two papers [25,26]. Some papers employed specialized methods, for example the Vascular Modelling Toolkit (VMTK) was used for vessel segmentation and centerline prediction by Yap et al. [27]. Similarly, Walsh et al. [2] used two Vaa3D plugins for neuron and vessel tracing: MOST Neuron Tracing [28] and APP2 [29].

A wide range of software was used to facilitate episcopic image analysis. Of the 20 software packages listed across all the reviewed papers, the most widely used were Amira (Thermofisher) (48 papers), OsiriX (Pixmeo) (24 papers) and Fiji/ ImageJ (18 papers) (Figure 5A). Amira was particularly popular for segmentation, while OsiriX was used more often for object detection. Commercial software continues to be preferred over open-source alternatives: there were 35 instances of open source software being used for image analysis, compared to 99 instances of commercial software use. 22 papers did not list the software used for their analysis. 

In three papers, deconvolution algorithms were used to overcome the image blurring caused by out-of-plane light from below the sample surface [2,26,30]. The asymmetric nature of this blurring necessitated a custom deconvolution algorithm in each case. Walsh et al. [30] presents a detailed pipeline for asymmetric deconvolution and demonstrates how this enables more accurate automated segmentation.

Asymmetric blurring is one of the reasons that automated methods might struggle to correctly annotate episcopic images. Additionally, where fluorescent labels are used, the fact that samples are whole-mount stained means they are likely to suffer from heterogenous levels of staining (due to poor penetration of the staining molecule into the center of the sample). The different forms of episcopic imaging also require different tools for annotation. This could explain the high proportion of papers relying on manual annotation, which represents a significant bottleneck for episcopic image analysis. This result highlights the need for new automated methods of annotation designed specifically for episcopic image data. One area of research that could address this need is future is machine learning-based image annotation.

## 4. Discussion

Publication of episcopic imaging data has increased over the last decade, demonstrating that episcopic microscopy continues to be a valuable tool for analyzing biological samples. Quantitative analysis of this data has also become more prevalent, with an increasing range of methodologies being used to extract quantitative data from episcopic images. However, the proportion of papers providing quantitative image analysis is still fairly low, averaging 54%, possibly due in part to the difficulties associated with image annotation: episcopic images are mostly annotated by hand, which can be extremely labor intensive. While some areas of analysis follow standardized methods (for example, fractal analysis of the heart and scoring of abnormalities according to an atlas) much of the quantification being performed is not standardized across research groups. 20 different software packages are used for analysis, most of which are commercially licensed and potentially expensive, while 22 papers did not list the software used. This makes reproducing and comparing results challenging, especially as different software packages provide different tools for visualization (for example, arbitrary reslicing) which can influence the manual segmentation produced. 

## 5. Future Directions

For quantitative episcopic imaging to continue to grow as a technique, future work must focus on automated image annotation and standardized pipelines for quantitative analysis. Given recent trends, it is likely we will see continue to see more use of complex or multi-stage methodologies, for example segmentations that feed in to computer simulations. In these cases, a reproducible analysis pipeline is even more crucial to provide confidence in the analysis results. Making data easily available, using open source software, and clearly defining the analysis pipeline used in a published study are all ways researchers can contribute to more standardized and reproducible quantification methodologies. One exciting avenue for further research is machine learning-based image annotation. Episcopic image data is a prime candidate for machine learning (ML) approaches due to the availability of data in online repositories that could be used for training. The development of new tools, such as ML models, for episcopic image analysis will expand the use of episcopic imaging and create further possibilities for new applications.

## Figures and Tables

**Figure 2 biomedicines-11-00909-f002:**
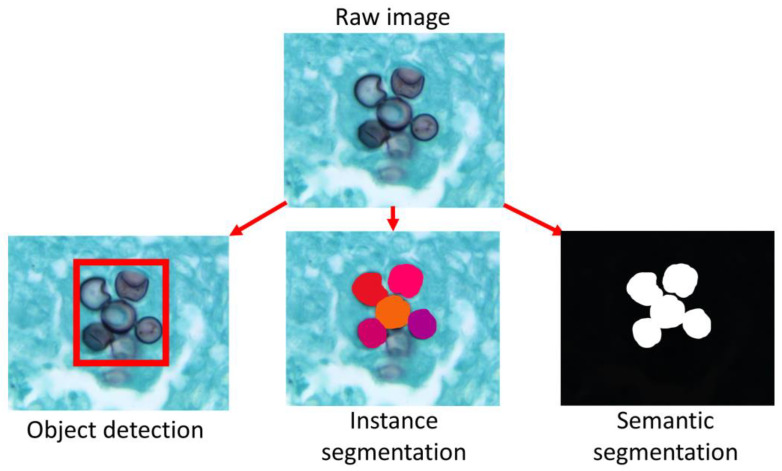
Here a cluster of *Paracoccidioides brasiliensis* cells are annotated in a photomicrograph (source: CDC Public Health Image Library) by three different methods: object detection, instance segmentation and semantic segmentation. Object detection results in a region of interest (red border, right panel) while instance segmentation produces a mask of the area of each cell (colored regions, center panel) and semantic segmentation produces a binary image with each pixel labelled as ‘background’ (black) or ‘cell’ (white) (left panel).

**Figure 3 biomedicines-11-00909-f003:**
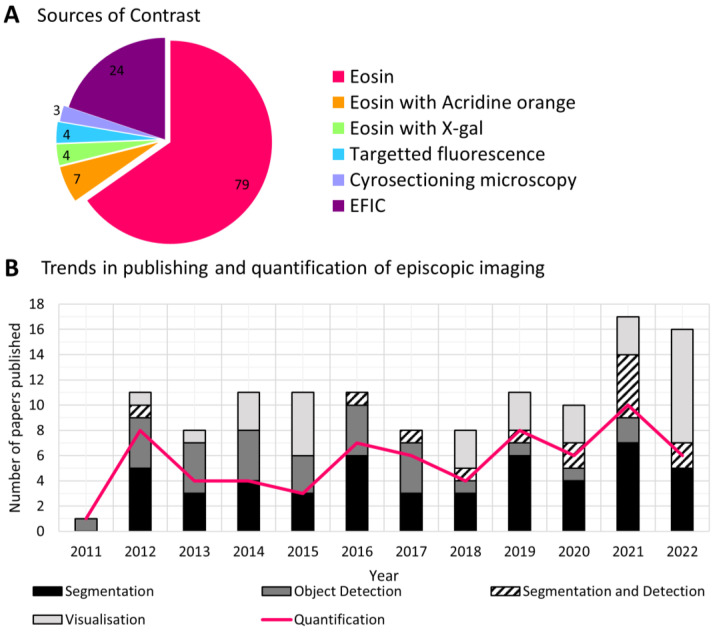
(**A**) The relative proportion of papers using each form of contrast is shown in a pie chart. Eosin contrast is by far the most widely used, followed by EFIC. (**B**) The trends in publishing of episcopic image data, the type of annotation performed, and the presence of quantification are summarized. The number of papers published per year has remained steady, but the proportion of those papers that include segmentation of episcopic image data has increased over time.

**Figure 4 biomedicines-11-00909-f004:**
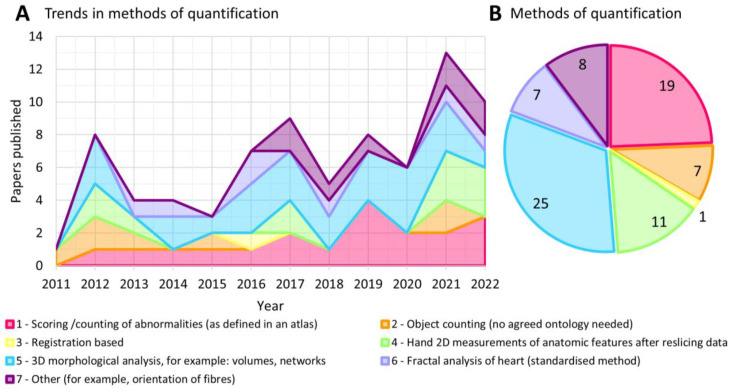
(**A**) This graph shows the trends in methods of image quantification over the last decade. The number of papers employing any method of quantification increased over time, with scoring of abnormalities (method 1) and 3D morphological analysis (method 5) showing the greatest increase. (**B**) The total number of papers using each quantification method since 2011 are shown in a pie chart, highlighting the scoring of abnormalities and 3D morphological analysis as the two most commonly used methods.

**Figure 5 biomedicines-11-00909-f005:**
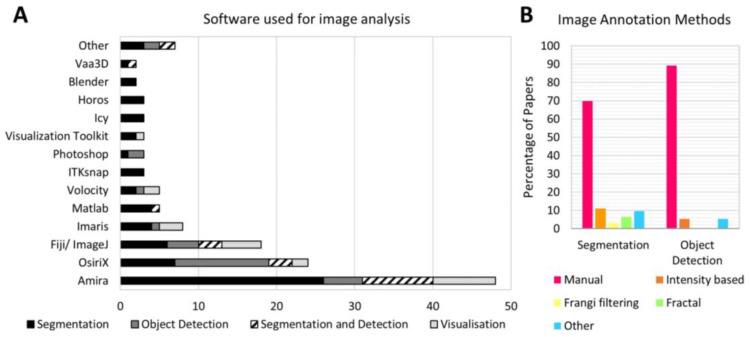
(**A**) Software used for image analysis varied depending on the type of image annotation being performed. Amira was the most popular software for segmentation while OsiriX was more popular for object detection. ImageJ/Fiji was the most commonly used open-source software. (**B**) The relative frequency of different annotation methods is shown in a bar chart, subdivided by annotation type (segmentation or object detection). Manual annotation was most common in both cases.

## Data Availability

An excel spreadsheet containing a summary of the published articles included in our meta-analysis, along with the data used to produce all figures within this review, is available for download via this link: https://github.com/natalie11/Systematic-review-of-HREM-quantiative-imaging (accessed on 14 March 2023).

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
