# Peer review of "Quantitative Image Processing for Three-Dimensional Episcopic Images of Biological Structures: Current State and Future Directions"

_biomedicines, 2023, doi:10.3390/biomedicines11030909_

Round 1

Reviewer 1 Report

Figure 2, and 4 are slightly blurred. Use high quality figures, if possible. 

Author Response

Dear Reviewer,

Thank you for taking the time to read our manuscript. We have taken on board you comments and replaced the figures with higher quality images. We have also added more references to relevant literature.

Sincerely,

Natalie Holroyd

Reviewer 2 Report

Holroyd et al present a meta-analysis of papers on episcopic imaging, principally microscope techniques, published between 2011 and 2022. After a short introduction they give a brief overview of annotation techniques and show that quantitative approaches have become more relevant in recent years.

I am no sure whether this paper in its current form is in the interest of the whole readership of Biomedicines. For that to be the case I would expect it to be a more detailed review that introduces the topic in more detail, for example through a figure in the introduction that shows some three-dimensional episcopic images obtained through the different techniques (HREM, EFIC, MF-HREM etc.) and compares them with conventional (fluorescence) microscope images. Together with a more detailed description/discussion of the techniques that could help to interest the general readership for the topic.

The results and discussion parts have nearly no “review character” in the way they are organized. The reader is basically informed about statistics (X papers used technique Y and performed annotation by method Z) and very few trends and conclusions that are relevant to a broader audience are presented a result of this. In my opinion, individual papers that are exemplary or outstanding for some reasons from the chosen 123 should be discussed in each of the subsections discussing the elected criteria. In that way the reader could see the full potential of episcopic imaging, its field of applications and the best techniques for its application. This could help people already in the field but also create interest in the more general readership and give them reasons and starting points for beginning their own experiments or collaborations in order to address a research question they have.

Additionally, I think the table with the 123 identified papers should at least be part of a Supporting Information that is directly linked to the article or parts of it should even be presented as overview tables for the subsections in the paper. This allows the reader to directly identify works of interest in their context and together with the description/discussion of the technique or method for image analysis. In connections with this I also don´t understand why there are only 26 references in the manuscript when the authors have identified 123 works that fall under the criteria of the review.

Overall, I think that the paper in its current form does not provide a review on “Quantitative image processing for three-dimensional episcopic images of biological structures: current state and future direction”. Instead it just hints at possible future directions based on trends identified by statistics. I therefore recommend major revisions.

Author Response

Dear Reviewer,

Thank you for taking the time to read our manuscript and provide feedback. We have taken on board you comments and have implemented changes that we feel greatly improve the manuscript.

  1. We have incorporated a table summarising all 123 reviewed papers in a Supplementary Materials document and added references to all papers in the bibliography.
  2. We have added a figure to illustrate the set up used for episcopic microscopy, along with three dimensional rendering of the data produced by each technique (HREM, EFIC, MF-HREM). We have included a short discussion of the benefits of episcopic imaging and why it might be used over alternatives such as histology, confocal or light sheet microscopy. While we envision this review being most useful to readers of the “Visualizing 3D Embryo and Tissue Morphology—A Decade of Using High-Resolution Episcopic Microscopy (HREM) in Biomedical Imaging, Volume II” special issue who are already using epsicopic techniques, we hope that this addition will broaden the reach of our paper and increase interest in these imaging methodologies.

  3. We have added discussions of specific papers to each section to highlight how different analysis techniques are being used in the literature. We make reference to the specific tool being used to perform analysis in these cases and show how episcopic image analysis contributed to the findings of each paper. A range of example papers were chosen to showcase the breadth of applications for with episcopic microscopy can be used.

    We believe these will give readers a better insight into how they might include quantitative analysis of episcopic images in their own work, while also highlighting the areas for further research.

    Sincerely,
    Natalie Holroyd  

Round 2

Reviewer 2 Report

I would like to thank the authors for their response and the changes they made. I recommend acceptance of the revised version.